# Insights on the vulnerability of Antarctic glaciers from the ISMIP6 ice sheet model ensemble and associated uncertainty

Hélène Seroussi [1], Vincent Verjans [2], Sophie Nowicki [3], Antony J. Payne [4], Heiko Goelzer [5], William H. Lipscomb [6], Ayako Abe-Ouchi [7], Cécile Agosta [8], Torsten Albrecht [9], Xylar Asay-Davis [10], Alice Barthel [10], Reinhard Calov [9], Richard Cullather [11], Christophe Dumas [8], Benjamin K. Galton-Fenzi [12, 13, 14], Rupert Gladstone [15], Nicholas R. Golledge [16], Jonathan M. Gregory [17,18], Ralf Greve [19,20], Tore Hattermann [21], Matthew J. Hoffman [10], Angelika Humbert [22,23], Philippe Huybrechts [24], Nicolas C. Jourdain [25], Thomas Kleiner [22], Eric Larour [26], Gunter R. Leguy [6], Daniel P. Lowry [27], Chistopher M. Little [28], Mathieu Morlighem [29], Frank Pattyn [30], Tyler Pelle [31], Stephen F. Price [10], Aurélien Quiquet [8,25], Ronja Reese [9, 32], Nicole-Jeanne Schlegel [33,26], Andrew Shepherd [32], Erika Simon [11], Robin S. Smith [17], Fiammetta Straneo [31], Sainan Sun [32], Luke D. Trusel [34], Jonas Van Breedam [24], Peter Van Katwyk [35], Roderik S. W. van de Wal [36,37], Ricarda Winkelmann [9,38], Chen Zhao [14], Tong Zhang [39], and Thomas Zwinger [40]

[1]Thayer School of Engineering, Dartmouth College, Hanover, NH, USA
[2]Center for Climate Physics, Institute for Basic Science, Busan, Republic of Korea
[3]University at Buffalo, Buffalo, NY, USA
[4]University of Bristol, United Kingdom
[5]NORCE Norwegian Research Centre, Bjerknes Centre for Climate Research, Bergen, Norway
[6]Climate and Global Dynamics Laboratory, National Center for Atmospheric Research, Boulder, CO, USA
[7]Atmosphere and Ocean Research Institute, University of Tokyo, Kashiwa, Japan
[8]Laboratoire des Sciences du Climat et de l'Environnement, LSCE-IPSL, CEA-CNRS-UVSQ, Université Paris-Saclay, Gif-sur-Yvette, France
[9]Potsdam Institute for Climate Impact Research (PIK), Member of the Leibniz Association, P.O. Box 60 12 03, 14412 Potsdam, Germany
[10]Fluid Dynamics and Solid Mechanics Group, Los Alamos National Laboratory, Los Alamos, NM, USA
[11]NASA Goddard Space Flight Center,Greenbelt, MD, USA
[12]Australian Antarctic Division, Kingston, Tasmania, Australia
[13]Australian Centre for Excellence in Antarctic Science, University of Tasmania, Hobart, Australia
[14]Australian Antarctic Program Partnership, Institute for Marine and Antarctic Studies, University of Tasmania, Hobart, Australia
[15]Arctic Centre, University of Lapland, Finland
[16]Antarctic Research Centre, Victoria University of Wellington, New Zealand
[17]National Centre for Atmospheric Science, University of Reading, United Kingdom
[18]Met Office Hadley Centre, Exeter, United Kingdom
[19]Institute of Low Temperature Science, Hokkaido University, Sapporo, Japan
[20]Arctic Research Center, Hokkaido University, Sapporo, Japan
[21]Norwegian Polar Institute, iC3: Centre for ice, Cryosphere, Carbon and Climate, Tromsø, Norway
[22]Alfred Wegener Institute for Polar and Marine Research, Bremerhaven, Germany
[23]Department of Geoscience, University of Bremen, Bremen, Germany
[24]Earth System Science and Departement Geografie, Vrije Universiteit Brussel, Brussels, Belgium
[25]Univ. Grenoble Alpes/CNRS/IRD/G-INP, Institut des Géosciences de l'Environnement, France
[26]Jet Propulsion Laboratory, California Institute of Technology, Pasadena, CA, USA

[27]GNS Science, Lower Hutt, New Zealand

[28]Atmospheric and Environmental Research, Inc., Lexington, Massachusetts, USA

[29]Department of Earth Sciences, Dartmouth College, Hanover, NH, USA

[30]Laboratoire de Glaciologie, Université Libre de Bruxelles, Brussels, Belgium

[31]Scripps Institution of Oceanography, University of California San Diego, La Jolla, CA, USA

[32]University of Northumbria, Newcastle upon Tyne, United Kingdom

[33]NOAA Geophysical Fluid Dynamics Laboratory, Princeton, NJ, USA

[34]Department of Geography, Pennsylvania State University, University Park, PA, USA

[35]Department of Earth, Environmental, and Planetary Sciences, Brown University, Providence, RI, USA

[36]Institute for Marine and Atmospheric research Utrecht, Utrecht University, The Netherlands

[37]Department of Physical Geography, Utrecht University, Utrecht, the Netherlands

[38]University of Potsdam, Institute of Physics and Astronomy, Karl-Liebknecht-Str. 24-25, 14476 Potsdam, Germany

[39]State Key Laboratory of Earth Surface Processes and Resource Ecology, Beijing Normal University, Beijing, China

[40]CSC-IT Center for Science, Espoo, Finland

**Correspondence:** Hélène Seroussi (helene.l.seroussi@dartmouth.edu)

**Abstract.** The Antarctic Ice Sheet represents the largest source of uncertainty in future sea level rise projections, with a contribution to sea level by 2100 ranging from -5 to 43 cm of sea level equivalent under high carbon emission scenarios estimated by the recent Ice Sheet Model Intercomparison for CMIP6 (ISMIP6). ISMIP6 highlighted the different behaviors of the East and West Antarctic ice sheets, as well as the possible role of increased surface mass balance in offsetting the dynamic ice loss in response to changing oceanic conditions in ice shelf cavities. However, the detailed contribution of individual glaciers, as well as the partitioning of uncertainty associated with this ensemble, have not yet been investigated. Here, we analyze the ISMIP6 results for high carbon emission scenarios, focusing on key glaciers around the Antarctic Ice Sheet, and we quantify their projected dynamic mass loss, defined here as mass loss through increased ice discharge into the ocean in response to changing oceanic conditions. We highlight glaciers contributing the most to sea level rise as well as their vulnerability to changes in oceanic conditions. We then investigate the different sources of uncertainty and their relative role in projections, for the entire continent and for key individual glaciers. We show that, in addition to Thwaites and Pine Island glaciers in West Antarctica, Totten and Moscow University glaciers in East Antarctica present comparable future dynamic mass loss and high sensitivity to ice shelf basal melt. The overall uncertainty in additional dynamic mass loss in response to changing oceanic conditions, compared to a scenario with constant oceanic conditions, is dominated by the choice of ice sheet model, accounting for 52% of the total uncertainty of the Antarctic dynamic mass loss in 2100. Its relative role for the most dynamic glaciers varies between 14% for MacAyeal and Whillans ice streams and 56% for Pine Island Glacier at the end of the century. The uncertainty associated with the choice of climate model increases over time and reaches 13% of the uncertainty by 2100 for the Antarctic Ice Sheet, but varies between 4% for Thwaites glacier and 53% for Whillans ice stream. The uncertainty associated with the ice-climate interaction, which captures different treatments of oceanic forcings such as the choice of melt parameterization, its calibration, and simulated ice shelf geometries, accounts for 22% of the uncertainty at the ice sheet scale, but reaches 36 and 39% for Institute ice stream and Thwaites Glacier, respectively, by 2100. Overall, this study helps inform future research by highlighting the sectors of the ice sheet most vulnerable to oceanic warming over the 21st century and by quantifying the main sources of uncertainty.

# 1 Introduction

Remote sensing observations show that the Antarctic Ice Sheet has lost the equivalent of 14 mm of sea level rise over the past four decades, and this trend is accelerating (Shepherd et al., 2018; Rignot et al., 2019; Hamlington et al., 2020). The long-term contribution of this ice sheet to sea level rise is the largest source of uncertainty in current projections, as estimates for 2100 range between -5 and 43 cm (IPCC, 2021; Edwards et al., 2021; van de Wal et al., 2022). The Ice Sheet Model Intercomparison for CMIP6 (ISMIP6, Nowicki et al., 2016, 2020) was designed to improve projections of ice sheet evolution over the coming century using an ensemble of state-of-the-art ice sheet models forced with nine different climate models. ISMIP6 simulations under the high carbon emission scenarios RCP8.5 and SSP5–8.5 showed that the Antarctic Ice Sheet could contribute between -8 and +30 cm of Sea Level Equivalent (SLE) on top of the current trend (Seroussi et al., 2019, 2020). Simulations suggest that most of the mass loss comes from the West Antarctic Ice Sheet (WAIS), with several basins vulnerable to sub-ice shelf ocean conditions by the end of the century (Naughten et al., 2018; Purich and England, 2021). However, the overall dynamic response of WAIS is compensated, at least partially, by a projected increase in snowfall, mostly in East Antarctica, as warmer air conditions allow the atmosphere to transport more moisture over the ice sheet (Huybrechts and Oerlemans, 1990; Seroussi et al., 2020). The analysis by Edwards et al. (2021), using a statistical emulator, further investigates the sensitivity of the results to climate scenarios. They showed a weak sensitivity of Antarctic projections to carbon emission scenario, unlike that which is observed for the Greenland Ice Sheet and other glaciated regions around the world, where ice loss is strongly influenced by the future climate scenario. A comparison of Antarctic projections using CMIP5 and CMIP6 forcings shows no significant difference in total Antarctic ice loss when using the previous (CMIP5) or newest (CMIP6) climate model forcings (Payne et al., 2021). The generally warmer conditions in CMIP6 cause both larger surface mass balance and warmer ocean conditions. While the former increases snowfall rates, the latter leads to more ice shelf thinning, inducing a loss of buttressing, and an increased ice discharge into the ocean (Dupont and Alley, 2005). Overall, these two driving processes compensate each other. All these previous studies shed light on the future Antarctic contribution to sea level but only addressed this contribution at large scale, without investigating the role and response of individual Antarctic glaciers. Furthermore, they attributed the uncertainty to climate forcings, ice flow models, and sub-ice shelf melt parameterizations but did not investigate the relative role of these sources of uncertainty, as well as their evolution over time.

Here, we investigate the role of individual glaciers in the overall Antarctic Ice Sheet contribution to sea level, and estimate their vulnerability to increases in ocean-induced melt at the base of the adjoining ice shelves. We quantify the role of the different sources of uncertainty on Antarctic mass loss, including climate forcing and ice flow models. We focus on the RCP8.5 and SSP5–8.5 scenarios only, as Edwards et al. (2021) demonstrated a limited sensitivity of the Antarctic Ice Sheet evolution by 2100 to carbon emission scenarios, and only a limited number of experiments were made with RCP2.6 and SSP1–2.6 as part of the ISMIP6 Antarctic model ensemble. The goal of this study is to investigate the dynamic mass loss of glaciers, and we isolate this signal in the overall mass change. We first summarize the ISMIP6 experiments included in this analysis, the forcings applied, and describe our methodology. We then quantify the role of individual glaciers' dynamic change to the

overall Antarctic sea level contribution and their sensitivity to varying climate conditions. We finally partition the uncertainty at continental and local scale, and summarize our conclusions and their impact on future research directions.

## 2 Data and Methods

### 2.1 Model ensemble

We use the ensemble of simulations from ISMIP6-Antarctica, as described in Seroussi et al. (2020) and Payne et al. (2021). Simulations were performed by 13 ice flow modeling groups with different ice flow models and are forced with ocean and atmospheric conditions from 9 different global climate and Earth system models, 6 CMIP5 models and 3 CMIP6 models, over the 2015–2100 period (see Table 1). We only consider here the high emission scenarios, RCP8.5 for CMIP5 models and SSP5–8.5 for CMIP6 models, which we consider to be similar in terms of overall forcings for our analysis. No other scenario is used, as the predicted Antarctic mass loss showed little sensitivity to carbon emission scenario (Edwards et al., 2021) and very few simulations were performed with other carbon emission scenarios as part of ISMIP6. In addition to these experiments with varying atmospheric and oceanic conditions, a control experiment (*ctrl_proj*) with constant climate conditions, similar to the 1980–2014 period, was performed (see Nowicki et al., 2020; Seroussi et al., 2020, for more information).

We also use another set of experiments with forcing from oceanic conditions only. In these experiments, surface mass balance remains unchanged over the simulation period, identical to that which is done in the *ctrl_proj* experiment. These additional experiments are run for three of the climate models; they were part of a lower tier of experiments and therefore have only been performed by a few ice flow models.

All the experiments and their main characteristics are listed in Table 1, while Table 2 provides the list of experiments performed by each ice flow model. In this study, we focus solely on the glaciers' dynamic response to changes in ocean conditions. Experiments including ice shelf collapse (Nowicki et al., 2020; Seroussi et al., 2020) are therefore not included in our analysis. All the experiments used are based on a medium ocean sensitivity (see Jourdain et al., 2020), and experiments with low or high ocean sensitivity are also excluded, as only a few simulations were performed using this setup. Simulations performed by various ice sheet models use different ice-shelf melt parameterizations, either based on the ISMIP6 proposed parameterizations (Jourdain et al., 2020) or other parameterizations described in the literature (Martin et al., 2011; DeConto and Pollard, 2016; Lazeroms et al., 2018; Reese et al., 2018; Pelle et al., 2020). The ISMIP6 parameterization estimates the melt at the base of ice shelves to vary quadratically as a function of thermal forcing; it calibrates the coefficient linking these terms as well as the temperature bias to match observed melt rates for each basin around Antarctica (Jourdain et al., 2020). Other parameterizations used by the models include the PICO box model that coarsely resolves the ocean overturning circulation in the ice shelf cavity and uses a boundary layer to compute the ice-shelf melt (Reese et al., 2018, 2020), or a buoyant meltwater plume model that evolves beneath the ice shelf based on the ice shelf geometry and plume properties (Lazeroms et al., 2018). The PICOP parameterization combines the PICO box model and a plume parameterization to both resolve the sub-shelf ocean circulation and capture the ice shelf geometry (Pelle et al., 2019). A quadratic melt parameterization with slope dependence was also used in which the ice shelf melt is additionally dependent on the local angle between the ice-shelf base and the horizontal

direction, as it tends to create larger melt close to grounding lines as suggested by remote-sensing estimates (Lipscomb et al., 2021). The choice of ice-shelf melt parameterization was left to the discretion of modeling groups, but all the parameterizations rely on the ocean conditions in ice-shelf cavities provided as part of ISMIP6 (see also table A1). The ISMIP6 parameterization is referred to as *standard* in our study, while all other parameterizations are referred to as *open* (see Table 1).

**Table 1.** List of ISMIP6–Antarctica experiments used with their main characteristics. Ice-shelf melt refers to the ice shelf parameterization used in ice flow models: "Standard" is the parameterization developed for ISMIP6 (see Jourdain et al., 2020) and "Open" is any other ice shelf melt parameterization.

| Experiment | Climate Model | Scenario | Ice-shelf melt | SMB forcing |
|---|---|---|---|---|
| ctrl_proj | None | None | Free | Constant |
| exp01 | NorESM1-M | RCP8.5 | Open | Varying |
| exp02 | MIROC-ESM-CHEM | RCP8.5 | Open | Varying |
| exp04 | CCSM4 | RCP8.5 | Open | Varying |
| exp05 | NorESM1-M | RCP8.5 | Standard | Varying |
| exp06 | MIROC-ESM-CHEM | RCP8.5 | Standard | Varying |
| exp08 | CCSM4 | RCP8.5 | Standard | Varying |
| expA1 | HadGEM2-ES | RCP8.5 | Open | Varying |
| expA2 | CSIRO-MK3 | RCP8.5 | Open | Varying |
| expA3 | IPSL-CM5A-MR | RCP8.5 | Open | Varying |
| expA5 | HadGEM2-ES | RCP8.5 | Standard | Varying |
| expA6 | CSIRO-MK3 | RCP8.5 | Standard | Varying |
| expA7 | IPSL-CM5A-MR | RCP8.5 | Standard | Varying |
| expB1 | CNRM-CM6 | SSP5–8.5 | Open | Varying |
| expB3 | UKESM-1 | SSP5–8.5 | Open | Varying |
| expB4 | CESM2 | SSP5–8.5 | Open | Varying |
| expB6 | CNRM-CM6 | SSP5–8.5 | Standard | Varying |
| expB8 | UKESM-1 | SSP5–8.5 | Standard | Varying |
| expB9 | CESM2 | SSP5–8.5 | Standard | Varying |
| expC2 | NorESM1-M | RCP8.5 | Open | Constant |
| expC3 | NorESM1-M | RCP8.5 | Standard | Constant |
| expC5 | MIROC-ESM-CHEM | RCP8.5 | Open | Constant |
| expC6 | MIROC-ESM-CHEM | RCP8.5 | Standard | Constant |
| expC11 | CCSM4 | RCP8.5 | Open | Constant |
| expC12 | CCSM4 | RCP8.5 | Standard | Constant |

    Previous ISMIP6 studies focused on large-scale changes of ice mass loss (Goelzer et al., 2020; Seroussi et al., 2020; Payne
et al., 2021; Edwards et al., 2021) and results have been processed to compute scalar quantities at continental scale and for the three main Antarctic regions: Antarctic Peninsula, West Antarctic Ice Sheet (WAIS), and East Antarctic Ice Sheet (EAIS). In order to assess local changes, we reprocessed the results at glacier scale using basins from the Ice sheet Mass Balance Inter-comparison Exercise (IMBIE; Shepherd et al., 2012). This dataset lists 198 individual glaciers of the Antarctic Ice Sheet, so we reprocessed the results to compute annual surface mass balance, ocean-induced basal melt, ice volume and ice volume

**Table 2.** List of experiments performed as part of ISMIP6-Antarctica by the different modeling groups and experiments emulated from these results. X: simulated experiments. *: emulated experiments (see section 2.3).

| Experiment | AWI_PISM | DOE_MALI | ILTS_PIK_SICOPOLIS | IMAU_IMAUICE1 | IMAU_IMAUICE2 | JPL1_ISSM | LSCE_GRISLI | NCAR_CISM | PIK_PISM1 | PIK_PISM2 | UCIJPL_ISSM | ULB_fETISh_16 | ULB_fETISh_32 | UTAS_ElmerIce | VUB_AISMPALEO | VUW_PISM |
|---|---|---|---|---|---|---|---|---|---|---|---|---|---|---|---|---|
| ctrl_proj | X | X | X | X | X | X | X | X | X | X | X | X | X | X | X | X |
| exp01 | X |  |  |  |  |  |  | X | X | X | X | X | X |  |  | X |
| exp02 | X |  |  |  |  |  |  | X | X | X | X | X | X |  |  | X |
| exp04 | X |  |  |  |  |  |  | X | X | X | X | X | X |  |  | X |
| exp05 | X | X | X | X | X | X | X | X |  |  | X | X | X | X | X |  |
| exp06 | X | X | X | X | X | X | X | X |  |  | X | X | X | X | X |  |
| exp08 | X | X | X | X | X | X | X | X |  |  | X | X | X | * | X |  |
| expA1 | X |  |  |  |  |  |  | X | * | * | * | X | X |  |  | * |
| expA2 | X |  |  |  |  |  |  | X | * | * | * | X | X |  |  | * |
| expA3 | X |  |  |  |  |  |  | X | * | * | * | X | X |  |  | * |
| expA5 | X | * | X | * | X | X | X | X |  |  | X | X | X | * | X |  |
| expA6 | X | * | X | * | X | X | X | X |  |  | X | X | X | * | X |  |
| expA7 | X | * | X | * | X | X | X | X |  |  | X | X | X | * | X |  |
| expB1 | X |  |  |  |  |  |  | X | * | * | * | * | * |  |  | * |
| expB3 | X |  |  |  |  |  |  | X | * | * | * | * | * |  |  | * |
| expB4 | X |  |  |  |  |  |  | X | * | * | * | * | * |  |  | * |
| expB6 | X | * | X | * | * | X | X | X |  |  | X | * | * | * | * |  |
| expB8 | X | * | X | * | * | X | X | X |  |  | X | * | * | * | * |  |
| expB9 | X | * | X | * | * | X | X | X |  |  | X | * | * | * | * |  |
| expC2 |  |  |  |  |  |  |  | X |  |  | X |  |  |  |  |  |
| expC3 |  | X |  |  |  | X | X | X |  |  | X |  |  |  |  |  |
| expC5 |  |  |  |  |  |  |  | X |  |  | X |  |  |  |  |  |
| expC6 |  | X |  |  |  | X | X | X |  |  | X |  |  |  |  |  |
| expC11 |  |  |  |  |  |  |  | X |  |  | X |  |  |  |  |  |
| expC12 |  | X |  |  |  | X | X | X |  |  | X |  |  |  |  |  |

above floatation for these 198 glacier basins individually. We consider the extent of the basins to remain fixed over the 85-year period of the simulations, as ice divides change slowly on this timescale. Similarly to previous ISMIP6 studies (Seroussi et al., 2020; Goelzer et al., 2020; Edwards et al., 2021; Payne et al., 2021), we aim to analyze the impact of changes in climate conditions, and results from the simulations are therefore presented as experiment minus control. This approach allows us to remove the trend of individual ice flow models, as these trends are not consistent with the recent trend captured by remote sensing observations (Aschwanden et al., 2021; Nowicki and Seroussi, 2018; Schlegel et al., 2018) and vary between ice flow simulations. Nevertheless, this approach neglects the imbalance of the glaciers and potential non-linearities between response

to climate forcing and initial trend in ice sheet evolution. Previous results suggest such non-linearities are relatively limited on this timescale (Seroussi et al., 2014; Nowicki et al., 2013a).

## 2.2 Dynamic mass loss

Changes in Antarctic ice mass are caused by both changes in surface mass balance (e.g., with increased precipitation or additional runoff) and dynamic mass change (e.g., in response to changes in ice shelf geometry, driving stress, etc). In this study, we focus on the impact of changes in oceanic conditions only. As most simulations include both varying atmospheric and oceanic conditions, we remove the anomalies in surface mass balance from the overall mass change to compute the dynamic response component of the Antarctic Ice Sheet mass change. Changes in surface mass balance over Antarctic glaciers

do not significantly impact their driving stress and ice dynamics over a period of 85 years, as shown by previous studies (e.g., Seroussi et al., 2014). Previous ensemble experiments, such as the SeaRISE assessment (Bindschadler et al., 2013; Nowicki et al., 2013a, b), also demonstrated that changes from combined external forcings could be reconstructed by combining the signals from experiments with independent forcings (Bindschadler et al., 2013). We therefore consider this approach to be a good first order approximation to isolate the dynamic changes from the overall mass change.

To test this assumption further, we compare results obtained using this approximation (i.e., subtracting changes in surface mass balance from the overall mass change) to experiments performed with changes in oceanic forcing only, referred to as 'ocean-calculated' and 'ocean-only' experiments, respectively. Five ice flow models performed a total of 21 ocean-only experiments that can be compared to corresponding ocean-calculated experiments (see Table 1 for corresponding experiments).

Figure 1 shows that the total ice mass change varies on average by 3.5 mm SLE or 1.1% of the volume change (between

0.6 and 11.2 mm SLE depending on the ice flow and climate models used, see Fig. 1a) between the ocean-only and ocean-calculated experiments in 2100. Similarly, differences in ice volume above floatation for these two cases vary on average by 3.1 mm SLE (between 0.0 and 10.0 mm SLE). These numbers can however represent up to 100% in the cases where changes are limited and the change in volume above floatation is close to 0 mm SLE (Fig. 1b). Changes in ocean-induced basal melt are similar, with less than 1.3% difference in all cases (Fig. 1c), while changes in surface mass balance (not shown) are also

similar in both types of experiments, with less than 0.2% difference. These results confirm that removing changes in surface mass balance from the overall results is a reasonable approximation to isolate the dynamic changes of the Antarctic Ice Sheet.

## 2.3 Emulation of results

Table 2 shows the 128 experiments performed by ice flow models (experiments *exp01 – expB9*). Similarly to previous studies (Edwards et al., 2019, 2021), we use a statistical emulator to recreate the missing simulations and limit the bias introduced

by the missing experiments in our analysis of the sources of uncertainty. We therefore complete the list of simulations by adding emulated results for the missing experiments: we emulate statistically missing simulations not performed by the ice sheet models. For any ice sheet model, we emulate experiments with a given melt parameterization (standard or open) only if there are existing results using the same parameterization.

To emulate these missing experiments, we use Gaussian Process Regression to estimate the dynamic sea level contribution of the Antarctic Ice Sheet every year over 2015–2100 (separate emulators are also used for individual glaciers, see section 2.4). Each year is treated independently and emulated separately. The predictor variables in our emulators are the ice flow model used, the spatially-averaged warming of the ocean over the continental shelf since 2015, and the basal melt parameterization (open or standard). Adding other predictors, such as model resolution, basal sliding law or calving parameterization did not improve the ability to reproduce the simulated dynamic sea level change. The 128 existing simulations are split into a training (70%), a validation (20%) and a test (10%) set. We randomly assign the simulations to each set, selecting simulations such that the relative fraction of each ice sheet model in each set is similar to the initial ensemble, and ensuring afterwards that the relative number of each climate model is also respected compared to the initial ensemble. Fig. 2 shows the simulated and emulated values for the test set; some simulations are well captured by the emulation while others have larger errors, similar to Edwards et al. (2021). These errors are captured by the confidence interval of the emulator; the root mean square error (RMSE) of the emulator for the test set in 2100 is 7.5 mm SLE, and 97% of the simulated values fall within the 95% confidence intervals of the emulator. Emulated simulations generally underestimate the dynamic contribution, with a mean relative bias of -23% in 2100 in the test set

Once the emulator is chosen and validated, the 61 missing experiments are emulated. Only missing experiments with a similar basal melt parameterization are emulated: for example the IMAU_IMAUICE2 model only submitted experiments with the standard ice shelf melt parameterization, so experiments based on the open melt parameterization (see Table 1) are not emulated, hence the missing experiments in Table 2. Figure 3 shows the ensemble of dynamic sea level contribution simulated and emulated over the 2015–2100 period.

## 2.4 Glaciers contributing most to dynamic sea level rise

Dynamic mass loss from Antarctica is not uniformly distributed over the different basins and glaciers. Analyzing the spatial distribution of dynamic mass changes indicates that 11 glaciers contribute most to the overall dynamic sea level contribution, based on the simulations performed with the 9 climate models. These glaciers are Thwaites, Pine Island, Getz, Kamb, MacAyeal, Moller, Whillans and Institute in West Antarctica, and Leopold and Astrid Coast, Moscow University, and Totten in East Antarctica (see Fig. 4 for locations). Figure 4 shows, for each of the 9 climate models, the total dynamic sea level contribution from the Antarctic Ice Sheet as well as the dynamic contribution from these 11 glaciers. The average combined contribution of these 11 glaciers to dynamic sea level rise varies between 15.5 and 56.3 mm for the different climate models in 2100, and represents between one and two thirds of the total Antarctic dynamic sea level (between 33.1 and 64.7%). We investigate the sensitivity to changes in ocean conditions as well as the relative importance of the different sources of uncertainties for these 11 glaciers in the remainder of this manuscript.

In addition to the emulation of the dynamic sea level contribution for the entire Antarctic Ice Sheet, we also emulate the individual dynamic sea level contributions from these 11 glaciers contributing most to sea level over the simulation period. An approach similar to the Antarctic emulation is adopted, and each basin is emulated separately using the average ocean warming since 2015 within the sector where the glacier terminates. The hyperparameters of the Gaussian Process Regression are chosen

to best reproduce the ensemble for each basin. RMSE is below 1.5 mm for all the glaciers (between 0.31 mm for Whillans ice stream and 1.4 mm for Totten glacier), except Thwaites glacier with an RMSE of 5.5 mm. Between 86% and 100% of the results fall within the 95% confidence level of the emulators, depending on the glacier.

## 3   Results

### 3.1   Sensitivity to ocean induced melt

The dynamic contribution of individual glaciers to sea level rise varies depending on the amount of ocean-induced melt with changing oceanic conditions, and how vulnerable these glaciers are to such changes in basal melt. We use simulated results (excluding the emulated results) from the 9 climate models to estimate this sensitivity for the 11 glaciers around the Antarctic Ice Sheet, selected based on their large dynamic response to external forcings (see section 2.4). We compute the additional cumulative ice shelf melt compared to the control experiment over the 2015–2100 period, as well as the additional cumulative change in mass above floatation compared to the control experiment. We then analyze the sensitivity of mass loss to basal melt. When multiple ice streams feed the same ice shelf (e.g., Ross or Filchner-Ronne ice shelves), we distribute the melt between the different ice streams based on the closest ice stream from each grid point on the ice shelf.

Figure 5 shows the sensitivity of dynamic mass loss to changes in total 2015–2100 ice shelf melt for the 11 glaciers and for all the simulations listed in Table 1. The additional melt in each ice shelf varies depending on the climate model, the ice flow model, and the ice shelf melt parameterization used. The additional melt reaches $5 \times 10^4$ Gt over the 85 year period for Institute ice stream, up to $4 \times 10^4$ Gt over the 85 year period for Kamb and Whillans ice streams, $2 \times 10^4$ Gt under Thwaites Glacier, and between $1 \times 10^4$ Gt and $2 \times 10^4$ Gt for Pine Island, Totten and Moscow University glaciers. HadGEM2 tends to provide the largest warming in ocean conditions, especially under the Ross Ice Shelf. NorESM and UKESM also show strong warming in most regions, which translates into large melt increase for all the glaciers studied. Glaciers in the Amundsen Sea experience substantially warmer ocean conditions, relative to other glaciers, in the simulations based on CCSM4. These results are similar to that which is shown in Fig. 4 for the average changes by glacier.

The relationship between the additional melt and the dynamic mass loss is relatively linear at glacier scale, similar to that which was shown at the basin scale in Seroussi et al. (2020), especially for glaciers that are most sensitive to changes in ocean conditions. The correlation coefficient varies between 90.8% for Thwaites Glacier and 17.0% for Leopold and Astrid Coast. Thwaites Glacier has the largest sensitivity to changes in basal melt under its ice shelf (2.4 mm SLE/1,000 Gt), followed by Pine Island (1.3 mm SLE/1,000 Gt) and Totten (1.2 mm SLE/1,000 Gt) glaciers, as well as Moscow University Glacier (0.9 mm SLE/1,000 Gt). Ice streams feeding Ross and Filchner-Ronne ice shelves, on the other hand, experience very large increases in basal melt (>10,000 Gt), spread over very large areas, so they show relatively little response to these changes (0.1-0.7 mm SLE/1,000 Gt), as these large melting create only limited changes in ice shelf buttressing.

## 3.2 Sources of uncertainty

The results described above highlight a large spread in dynamic response, with climate models causing various changes in ocean conditions, and individual ice flow models responding differently to these changes. In order to quantify the role of the different sources of uncertainty, we use analysis of variance (ANOVA) theory (Girden, 1992; von Storch and Zwiers, 1999; Deque et al., 2007; Yip et al., 2011). This statistical technique allows us to analyze and compare the means of two or more groups to determine whether there are statistically significant differences between the groups. The total variability in the data is broken down into two components: the variance within each group (within-group variance) and the variance between the groups (between-group variance). The significance of differences in group means is evaluated based on the ratio of these two variance components. This approach allows us to decompose the variance of our modeled sea level contribution into the contribution of different components and the interactions between these different components. We partition the total uncertainty between contributions from climate models, ice flow models, and the interaction between these two terms (referred to as ice-climate interaction), similar to what was done by previous studies on mountain glaciers (Marzeion et al., 2020) or firn models (Verjans et al., 2021). The contribution from the ice flow models stems from all the differences that would arise between ice flow simulations even if the climatic forcing was identical, such as model resolution, sliding laws, calving, stress balance approximation, melt parameterization, etc. The ice-climate interaction term is the cross-term caused by interactions between the ice model and climate model variables, thus providing insight into the non-linearties between climate forcing and ice sheet response. It captures all the model parameters and conditions related to the ice-ocean interaction, including the choice of sub-ice shelf melt parameterization, its calibration, the geometry of the ice shelves, and the sensitivity of ice flow response to changes in buttressing. Consistent with the above analysis, we focus only on the role of ocean changes on the dynamic mass loss. We only include RCP8.5 and SSP5–8.5 scenarios, and since we consider these two scenarios to be similar, we do not distinguish between CMIP5 and CMIP6 in our experiments (Payne et al., 2021).

We account for the uncertainty inherent to emulator predictions by generating an ensemble of 100 dynamic mass loss changes for each combination of ice and climate models. We draw 100 values from the Gaussian distribution characterized by the predicted mean and standard deviation of the emulator. As discussed in section 2.3, the emulator properly captures emulation uncertainty (97% of the simulated test values fall into the 95% uncertainty intervals of the emulator), so the emulator uncertainty term is representative of errors introduced by substituting the emulator for ice sheet models in our experiments. For the existing simulations, we use the simulated values, which are therefore identical for the 100 ensemble members.

Figure 6 presents the evolution of the uncertainty over time for the Antarctic simulation ensemble as well as the relative contribution of different sources of uncertainty. Figure 6a shows the uncertainty associated with the ice models, the climate models, the ice-climate interaction, the emulation (including the contribution from the emulator only as well as the emulator-ice, emulator-climate, and emulator-ice-climate interaction terms), and the total standard deviation over the 2015–2100 period (see Table 3). The overall uncertainty in Antarctic evolution increases over time to reach 45 mm SLE by 2100, and the individual sources of uncertainty all increase over this time period. The uncertainty associated with the ice model choice dominates throughout the simulation period, followed by the uncertainty in ice-climate interactions. The emulation and climate models

uncertainty terms are lower and similar to each other. The overall uncertainty is very large and comparable to the mean signal of the projections.

Figure 6b shows the variance contributions from the different sources of uncertainty over time and Table 3 summarizes these results in 2100. The uncertainty associated with the choice of ice model varies between 45 and 80% over the period. The relative weight of the ice model changes during the first 25 years due to rapid adjustments to initial conditions in some ice flow models, but remains relatively stable at around 55% afterwards, with a small decrease over time. The relative role of the climate forcing on the other hand increases regularly from almost zero in 2015 to 13% of the total variance by the end of the century. The ice model and climate forcings have a strong interaction term (purple ice-climate area on Fig.6b), with the variance from ice-climate accounting for about 20% of the total variance throughout the simulations. The variance of the emulation term, which captures all the terms introduced by the emulator, is similar to the climate variance. This term represents 12 to 25% of the uncertainty, with rapid changes in the first 25 years and a stabilization at around 13% afterwards.

Figure 7 shows the overall uncertainty and its different components for the 11 individual glaciers selected in section 2.4, and Table 3 summarizes these number for 2100. The total uncertainty for Thwaites Glacier reaches 18.7 mm SLE (for a mean projected sea level of 7.7 mm SLE) in 2100, compared to a total Antarctic uncertainty of 45.3 mm (for a mean Antarctic sea level projection of 46.5 mm SLE). It is dominated by uncertainty from the ice flow models, followed closely by the ice-climate interaction, while the contribution of the other terms is limited. For all the other glaciers, the uncertainty by 2100 varies between 1.3 mm SLE for Whillans ice stream and 5.7 mm SLE for Institute ice stream. Uncertainty associated with different ice flow models dominates most glaciers in addition to Thwaites: Pine Island, Totten, Getz, Moscow as well as Leopold and Astrid Coast, similar to the Antarctic-wide uncertainty. However, uncertainty from climate forcing dominates for three ice streams of the Siple Coast, the ice-climate interaction dominates for Institute ice stream, and all three sources are comparable for Möller ice stream glaciers. The three ice streams on Siple Coast exhibit this differing behavior because of the large variation in ocean conditions simulated in the Ross ice shelf cavity by different climate models: several simulate very little change in ocean conditions in these cavities, while a few others predict substantial warming by the end of the century (Nowicki et al., 2020; Jourdain et al., 2020), causing a very large spread of ocean conditions and, therefore, sub-ice shelf basal melt.

Figure 8 shows the relative variance for the different glaciers and the different sources of uncertainty. The relative variance caused by climate conditions increases over time for all the glaciers; by 2100 it represents between 4% of the overall variance for Thwaites Glacier and 53% for Whillans ice stream. The uncertainty introduced by the emulation remains below 11% for Thwaites Glacier, but can reach 32% for Getz (ignoring the first simulation year that has very large values).

# 4 Discussion

In this study, we investigate the dynamic response of the Antarctic Ice Sheet to changes in oceanic conditions. We selected 11 glaciers, including 8 in West Antarctica and 3 in East Antarctica, that respond most to changes in oceanic conditions and contribute 33.1–64.7% of the additional sea level rise compared to simulations with constant climate conditions (see table 3), consistent with previous dynamic contributions found by Golledge et al. (2019). These glaciers all experience a relatively

linear relationship between dynamic sea level contribution and sub-ice shelf melt anomaly, which is consistent with findings at basin scale by Levermann et al. (2020) in the Linear Antarctic Response Model Intercomparison Project (LARMIP). Thwaites glacier is the most sensitive to additional basal melt, followed by Pine Island and Totten glaciers, as well as Moscow University glacier (Fig. 5). Monitoring and understanding oceanic conditions in the vicinity of the ice shelves of these four glaciers is therefore critical, as simulations suggest that they are not only very sensitive to changes (Fig. 5) but will also contribute a significant portion of the Antarctic dynamic mass loss by 2100 (Fig. 4). This is consistent with results from Schlegel et al. (2018) highlighting these glaciers as well as the glaciers ending in the Ronne ice shelf as the most sensitive to increased melt. Partitioning of the uncertainty in Antarctic dynamic mass loss projections shows that 55% of the overall variance by 2100 can be attributed to the simulation of ice dynamics, which is the largest source of uncertainty, and here represented by the choice of the ice sheet model. Due to the relatively limited number of ice flow simulations, we cannot assess which model parameters or processes contribute the most to such changes. At regional scale, we find that the uncertainty can be dominated by ice flow models (e.g., for Thwaites and Pine Island glaciers), the forcing from climate models (e.g., for MacAyeal and Whillans ice streams) or the interaction between ice and climate, which includes all the parameters and conditions related to the ice-ocean interaction (e.g., for Möller ice stream). Uncertainty is dominated by climate forcing for glaciers flowing into cold ice shelf cavities, and for which a subset of climate models simulates a large warming, as is the case for the Ross and Filchner-Ronne ice shelves.

**Table 3.** Summary table of mean sea level contribution, associated uncertainty, and relative variance of the climate models, ice sheet models, ice-climate interaction and emulator to the total variance in 2100. Values are listed for the Antarctic ice sheet as well as the 11 glaciers studied.

| Glacier name | Mean dynamic sea level rise (mm SLE) | Sea level uncertainty (mm SLE) | Climate model contribution | Ice sheet model contribution | Ice-climate contribution | Emulator contribution |
|---|---|---|---|---|---|---|
| Antarctic ice sheet | 46.5 | ±45.3 | 13% | 52% | 22% | 13% |
| Thwaites | 7.7 | ±18.7 | 4% | 49% | 39% | 8% |
| Pine Island | 2.5 | ±3.7 | 5% | 56% | 22% | 17% |
| Totten | 3.1 | ±3.5 | 24% | 36% | 28% | 11% |
| Getz | 1.8 | ±1.8 | 11% | 37% | 26% | 26% |
| Kamb | 1.4 | ±1.9 | 37% | 23% | 28% | 12% |
| MacAyeal | 1.3 | ±1.6 | 49% | 14% | 26% | 11% |
| Institute | 2.1 | ±5.7 | 29% | 26% | 36% | 9% |
| Moller | 1.3 | ±2.7 | 32% | 24% | 33% | 11% |
| Leopold and Astrid Coast | 1.1 | ±1.7 | 9% | 49% | 28% | 14% |
| Moscow | 2.1 | ±2.8 | 18% | 39% | 29% | 14% |
| Whillans | 1.2 | ±1.3 | 53% | 14% | 20% | 13% |

The main source of uncertainty for the Antarctic dynamic mass loss differs with the finding from a similar study applied to mountain glaciers (Marzeion et al., 2020). Overall, they found that the uncertainty is dominated by the carbon emission scenario by 2100. While our experiments do not consider the difference between carbon emission scenarios, as most ISMIP6

experiments are based on RCP8.5 and SSP5–8.5 forcings and only a few RCP2.6 and SSP1–2.6 forcings were conducted for
ISMIP6, Edwards et al. (2021) suggest that the the carbon emission scenario has a limited impact on the evolution of the
Antarctic ice sheet by 2100. While the results from Edwards et al. (2021) were performed on the overall mass loss of the
Antarctic Ice Sheet, including both the surface mass balance and the dynamic mass loss, here we assess only its dynamic
response to climate change. Since the dynamic mass loss and the surface mass balance generally compensate each other, and
surface mass balance is sensitive to the carbon emission scenario, it is likely that the Antarctic dynamic response is as well. We
also cannot discount that individual Antarctic glaciers could be more sensitive to emission scenario than the overall ice sheet.
Similarly to Marzeion et al. (2020), we observe that the relative impact of the choice of ice sheet models on the overall response
decreases over time, while the relative uncertainty associated with climate forcing and ice-climate interactions increases over
time. The ice flow models and climate models are independent, but they interact through the sensitivity of ice dynamics to
climate forcing. As such, the ice-climate interaction term in the uncertainty analysis shows non-negligible contributions to the
variance of about 20% at the Antarctic scale (Fig. 6b), and more than 40% for some glaciers in 2100, with an especially large
contribution for Thwaites Glacier. These ice-climate interaction terms are caused by the different treatment of oceanic forcings
by ice flow models: ice shelf geometries and ocean-induced melt parameterizations vary in the different ice flow models, and
this process creates interactions between the ice flow models and climate forcing. The uncertainty associated with the ice-
climate interaction term is therefore caused by the ice-ocean interactions and how the different ice sheet models respond to
given oceanic conditions.

The present study analyzes the dynamic mass loss of the Antarctic Ice Sheet and therefore does not include the sea level
contribution caused by changes in surface mass balance. Antarctic glaciers show a relatively limited dynamic response to
changes in surface mass balance on decadal timescales (Seroussi et al., 2014), and ice flow models respond relatively similarly
to such changes (Seroussi et al., 2019). Furthermore, previous results from the SeaRISE ensemble experiments showed the
relative linearity of responses to external forcings: adding mass loss from independent external forcings produced mass loss
similar to experiments combining these different external forcings together (Bindschadler et al., 2013). Removing the additional
surface mass balance over grounded ice from the total volume above floatation change is a good first order approximation to
capturing only the effect of oceanic changes, but differences exist between these results and those from simulations with
only ocean forcing, especially for simulations with limited contributions to sea level change (Fig.1). We do not expect these
differences to impact the overall results and partitioning of uncertainty presented in this study, but they would impact the exact
partitioning of uncertainty between the different components. Including surface mass balance to assess the overall uncertainty
in the total Antarctic mass loss instead of the dynamic component is beyond the scope of this study, but we expect that it would
increase the relative impact of climate models in the mass loss uncertainty. As seen in Seroussi et al. (2020), the total surface
mass balance anomaly over the grounded part of the Antarctic ice sheet varies between between 17 and 87 mm SLE over
the simulation period, which represents 17 to 217% of the mean dynamic loss found in this study. However, ice flow models
respond quite similarly to changes in surface mass balance (Seroussi et al., 2019).

Only a subset of ice flow models taking part in ISMIP6 Antarctica provided simulations for all the experiments listed in
Table 1, and a number of combinations of ice flow models and experiments therefore do not exist, causing some gaps in the

results. Therefore, as done in previous studies (Edwards et al., 2019, 2021), we use a statistical emulator to recreate some of the missing simulations and limit the bias introduced by these missing experiments. The uncertainty introduced by this emulator is not negligible, but remains limited to about 10-20% of the overall uncertainty. It can amount to more than 30% of the variance during certain time intervals, especially during transitions between ice- and climate-dominated uncertainties. This uncertainty could potentially be further reduced by investigating other emulator architectures, particularly those that enable more direct modeling of the time domain (Corani et al., 2021; Chantry et al., 2021). Due to computational constraints, the current emulator models projection years independently, which could introduce additional variance in emulator results between simulation years (Liu et al., 2020). Emulators based on neural networks are a potential solution for overcoming the existing computational constraints and directly modeling temporal data, as it has been shown that by efficiently handling larger amounts of data, neural network emulators are accurate and maintain the ability to quantify uncertainty (Van Katwyk et al., submitted). Another limitation comes from our treatment of the large ice shelves fed by several ice streams. We partition the overall ice shelf melt between the different ice streams based on the distance to the closest ice stream. The actual interplay between ice shelf thinning and grounded ice response is more complex (Fürst et al., 2016; Reese et al., 2017), and this separation of ice shelf melt is only a first-order approximation.

Results presented here are based on the experiments-minus-control approach used in Seroussi et al. (2020); Goelzer et al. (2020); Payne et al. (2021); Edwards et al. (2021), and therefore do not include the uncertainty associated with the control experiment. Analysis repeated for the experiments without subtracting the control run (while still subtracting changes caused by the surface mass balance) shows that in this case, the signal is entirely dominated by the ice flow model at continental scale (see Fig. 9). During the first 10 years, the total uncertainty is small and the emulation term accounts for at least 50% of the relative variance. The uncertainty grows very rapidly and becomes dominated by the ice flow model after 15 years. It accounts for 70 to 90% of the uncertainty during the last 70 years of the simulations, while it accounts for about 50% of the uncertainty when the trend from the control experiment is removed. This strong increase in ice model-related uncertainty is caused by between-model differences in trajectories of ice sheet dynamic sea level contributions at the start of the intercomparison experiment (i.e., 2015). The relative uncertainty caused by the climate and ice-ocean interactions both increase over time, but remain limited to less than 7% of the total variance. This suggests that continued improvements in ice flow models to better reproduce observed changes and validation of historical simulations with observations remain critical (Aschwanden et al., 2021). New developments to use transient assimilation instead of snapshot data assimilation (e.g., Goldberg et al., 2015; Larour et al., 2014) or to select simulations that best fit observations from a large ensemble with varying model parameters (e.g., DeConto and Pollard, 2016) offer opportunities to better constrain historical simulations but remain difficult to apply at large scale. Physical processes included in ice flow models will continue to evolve, and the number of parameterizations developed (e.g., sub-ice shelf melt, calving laws, sliding laws) is rapidly growing, which may further increase uncertainty over time, as models make different choices. Continued improvements in model development, verification and validation are therefore needed to reduce uncertainties in future dynamic mass loss, as they remain the main sources of uncertainty in the Antarctic Ice Sheet projections. Previous intercomparison efforts such as EISMINT (Payne et al., 2000), ISMIP-HOM (Pattyn et al., 2008), or MISMIP (Pattyn et al., 2012, 2013; Cornford et al., 2020) have allowed us to gain confidence in model numerical

implementations by comparing models against analytical solutions and against each other, and such efforts should continue to
help further reduce these uncertainties.

## 5   Conclusions

In this study we investigate the dynamic vulnerability of major Antarctic glaciers over the coming century under high carbon emission scenarios using the ISMIP6 ensemble of ice flow simulations. Specifically, we focus on the dynamic mass loss in response to changes in oceanic conditions. Our results show that, in addition to Thwaites and Pine Island glaciers in West Antarctica, Totten and Moscow University glaciers in East Antarctica both have the potential to respond rapidly to changes in oceanic conditions and can contribute significantly to sea level change by 2100. Uncertainty in future ice sheet dynamic contribution to sea level comes predominantly from the choice of ice flow model. This holds both at continental scale and for the individual Antarctic glaciers projected to contribute most to sea level change, even though the contribution of climate model choice to total uncertainty increases steadily throughout the century. Ice streams feeding the Ross ice shelf start to be mostly influenced by the choice of climate model in the 2050s, as some climate models produce large warming in these cold ice shelf cavities by the end of the century, while others suggest limited change in oceanic conditions. The ice flow models and climate models are independent, but they interact through the sensitivity of ice dynamics to climate forcing caused by the different treatment of oceanic forcings, including the choice of sub-ice shelf melt parameterizations, its calibration, and the simulated ice shelf geometries. When looking at the overall dynamic mass loss without subtracting the model trend under constant climate conditions, uncertainty is systematically dominated by the choice of ice flow models, highlighting the need for ice flow models to better capture and reproduce recent observed changes.

*Code and data availability.*  Code and data used to prepare this manuscript are permanently available on Zenodo with digital object identifyer: https://doi.org/10.5281/zenodo.8117513. Original ISMIP6 Antarctic Projection data are available on Zenodo for processed scalar data (https://doi.org/10.5281/zenodo.3940766) and on GHhub for 2 dimensional fields (https://theghub.org/dataset-listing).

## Appendix A:  Main characteristics of ice flow simulations

The table below summarizes the main characteristics of the ice flow models used as part the ISMIP6 Antarctica ensemble (Seroussi et al., 2020). It includes for each sets of simulations the numerics, stress balance approximation, model resolution, initialization method, initialization year, melt parameterization in partially floating cells, ice front parameterization, open and standard ice shelf melt parameterizations.

*Author contributions.*  H. Seroussi and S. Nowicki designed the study. H. Seroussi performed the analysis with help from V. Verjans and P. Van Katwyk. S. Nowicki, A.J. Payne, E. Larour, H. Goelzer, H. Seroussi, W.H. Lipscomb, A. Abe-Ouchi, J.M. Gregory, and A. Shepherd,

**Table A1.** List of ISMIP6-Antarctica Projections simulations and main model characteristics. Numerics: Finite Differences (FD), Finite Elements (FE), and Finite Volumes (FV). Initialization methods used: Spin-up (SP), Spin-up with ice thickness target values (SP+, see Pollard and DeConto, 2012), Data Assimilation (DA), Data Assimilation with relaxation (DA+), Data Assimilation of ice geometry only (DA*), and Equilibrium state (Eq). Melt in partially floating cells: Melt either applied or not over the entire cell based on a floating condition (Floating condition), melt applied based on a sub-grid scheme (Sub-grid), and N/A refers to models that do not have partially floating cells. Ice front migration schemes based on: strain rate (StR, Albrecht and Levermann, 2012), retreat only (RO), fixed front (Fix), minimum thickness height (MH) and divergence and accumulated damage (Div, Pollard et al., 2015). Basal melt rate parameterization in open framework: linear function of thermal forcing (Lin, Martin et al., 2011), quadratic local function of thermal forcing (Quad, DeConto and Pollard, 2016), PICO parameterization (PICO, Reese et al., 2018), PICOP parameterization (PICOP, Pelle et al., 2019), plume model (Plume, Lazeroms et al., 2018), and Non-Local parameterization with slope dependence of the melt (Non-Local + Slope, Lipscomb et al., 2021). Basal melt rate parameterization in standard framework: Local or Non-Local quadratic function of thermal forcing, Local or Non-Local anomalies (Jourdain et al., 2020).

| Model name | Numerics | Stress balance | Resolution (km) | Init. Method | Initial Year | Melt in partially floating cells | Ice Front | Open melt parameterization | Standard melt parameterization |
|---|---|---|---|---|---|---|---|---|---|
| AWI_PISM | FD | Hybrid | 8 | Eq | 2005 | Sub-Grid | StR | Quad | Non-Local |
| DOE_MALI | FE/FV | HO | 2-20 | DA+ | 2015 | Floating condition | Fix | N/A | Non-Local anom. |
| ILTS_PIK_SICOPOLIS | FD | Hybrid | 8 | SP+ | 1990 | Floating condition | MH | N/A | Non-Local |
| IMAU_IMAUICE1 | FD | Hybrid | 32 | Eq | 1978 | No | Fix | N/A | Local anom. |
| IMAU_IMAUICE2 | FD | Hybrid | 32 | SP | 1978 | No | Fix | N/A | Local anom. |
| JPL1_ISSM | FE | SSA | 2-50 | DA | 2007 | Sub-Grid | Fix | N/A | Non-Local |
| LSCE_GRISLI | FD | Hybrid | 16 | SP+ | 1995 | N/A | MH | N/A | Non-Local |
| NCAR_CISM | FE/FV | L1L2 | 4 | SP+ | 1995 | Sub-Grid | RO | Non-Local + Slope | Non-Local |
| PIK_PISM1 | FD | Hybrid | 8 | SP | 1850 | Sub-Grid | StR | PICO | N/A |
| PIK_PISM2 | FD | Hybrid | 8 | SP | 2015 | Sub-Grid | StR | PICO | N/A |
| UCIJPL_ISSM | FE | HO | 3-50 | DA | 2007 | Sub-Grid | Fix | PICOP | Non-Local |
| ULB_FETISH_16km | FD | Hybrid | 16 | DA* | 2005 | N/A | Div | Plume | Non-Local |
| ULB_FETISH_32km | FD | Hybrid | 32 | DA* | 2005 | N/A | Div | Plume | Non-Local |
| UTAS_ElmerIce | FE | Stokes | 4-40 | DA | 2015 | Sub-Grid | Fix | N/A | Local |
| VUB_AISMPALEO | FD | SIA+SSA | 20 | SP | 2000 | N/A | MH | N/A | Non-Local anom. |
| VUW_PISM | FD | Hybrid | 16 | SP | 2015 | No | StR | Lin | N/A |

designed the protocol of and lead ISMIP6. C. Agosta, X. Asay-Davis, A. Barthel, R. Cullather, T. Hattermann, N.C. Jourdain, C.M. Little, M. Morlighem, E. Simon, R.S. Smith, F. Straneo, and L.D. Trusel, derived the external forcing for the experiments. A.J. Payne, W.H. Lipscomb, T. Albrecht, R. Calov, C. Dumas, B.K. Galton-Fenzi, R. Gladstone, H. Goelzer, N.R. Golledge, R. Greve, M.J. Hoffman, A. Humbert, P.
Huybrechts, T. Kleiner, G.R. Leguy, D.P.Lowry, M. Morlighem, F. Pattyn, T. Pelle, S.F. Price, A. Quiquet, R. Reese, N.-J. Schlegel, H. Seroussi, S. Sun, J. Van Breedam, R.S.W. van de Wal, R. Winkelmann, C. Zhao, T. Zhang, and T. Zwinger performed the ice flow model simulations. H. Seroussi wrote the manuscript with inputs and edits from all authors.

*Competing interests.*  B.K. Galton-Fenzi and N.C. Jourdain are members of the editorial board of The Cryosphere.

*Acknowledgements.*  We thank Mira Berdahl and Nathan Urban for discussions on emulation and variance analysis. We thank the Climate and
Cryosphere (CliC) effort, which provided support for ISMIP6 through sponsoring of workshops, hosting the ISMIP6 website and wiki, and
promoted ISMIP6. We acknowledge the World Climate Research Programme, which, through its Working Group on Coupled Modelling,
coordinated and promoted CMIP5 and CMIP6. We thank the climate modeling groups for producing and making available their model
output, the Earth System Grid Federation (ESGF) for archiving the CMIP data and providing access, the University at Buffalo for ISMIP6
data distribution and upload, and the multiple funding agencies who support CMIP5 and CMIP6 and ESGF. We thank the ISMIP6 steering
committee, the ISMIP6 model selection group and ISMIP6 dataset preparation group for their continuous engagement in defining ISMIP6.
Helene Seroussi was supported by grants from the NASA Sea Level Change Team and Cryospheric Science programs (#80NSSC21K1939
and #80NSSC22K0383). Research was carried out by Nicole Schlegel and Eric Larour at the Jet Propulsion Laboratory, California Institute
of Technology, under a contract with the National Aeronautics and Space Administration. Sophie Nowicki was supported by NASA Sea
Level Change Team and Cryospheric Science programs (#80NSSC21K0915 and #80NSSC21K0322). Support for Xylar Asay-Davis, Alice
Barthel, Matthew Hoffman, and Stephen Price was provided by the Scientific Discovery Through Advanced Computing and Earth System
Model Development programs, funded by the U.S. Department of Energy, Office of Science. MALI simulations were performed on machines
at the National Energy Research Scientific Computing Center, a U.S. Department of Energy Office of Science User Facility located at
Lawrence Berkeley National Laboratory, operated under Contract No. DE-AC02-05CH11231. Rupert Gladstone and Thomas Zwinger were
supported by Academy of Finland grant nos. 322430 and 286587 and wish to acknowledge CSC – IT Center for Science, Finland, for
computational resources. Chen Zhao and Ben Galton-Fenzi received grant funding from the Australian Government as part of the Antarctic
Science Collaboration Initiative program (ASCI000002; Australian Antarctic Program Partnership). Ralf Greve was supported by Japan
Society for the Promotion of Science (JSPS) KAKENHI Grant Nos. JP16H02224, JP17H06104 and JP17H06323. Gunter Leguy and William
Lipscomb were supported by the National Center for Atmospheric Research, which is a major facility sponsored by the National Science
Foundation under Cooperative Agreement no. 1852977. Computing and data storage resources for CISM simulations, including the Cheyenne
supercomputer (https://doi.org/10.5065/D6RX99HX), were provided by the Computational and Information Systems Laboratory (CISL)
at NCAR. The work of Thomas Kleiner has been conducted in the framework of the PalMod project (FKZ: 01LP1511B), supported by
the German Federal Ministry of Education and Research (BMBF) as part of the Research for Sustainability initiative (FONA). Torsten
Albrecht and Ricarda Winkelmann were supported by the Deutsche Forschungsgemeinschaft (DFG) in the framework of the priority program
"Antarctic Research with comparative investigations in Arctic ice areas" by grants WI4556/2-1 and WI4556/4-1, and within the framework
of the PalMod project (FKZ: 01LP1925D) supported by the German Federal Ministry of Education and Research (BMBF) as a Research
for Sustainability initiative (FONA). Ronja Reese was supported by the Deutsche Forschungsgemeinschaft (DFG) by grant WI4556/3-1
and through the TiPACCs project that received funding from the European Union's Horizon 2020 Research and Innovation program under
grant agreement no. 820575. Development of PISM is supported by NASA grants 20-CRYO2020-0052 and 80NSSC22K0274 and NSF
grant OAC-2118285. Heiko Goelzer received funding from the European Union's Horizon 2020 Research and Innovation Programme under
grant agreement no. 869304, and used resources provided by Sigma2 - the National Infrastructure for High Performance Computing and
Data Storage in Norway through projects NS5011K, NN8085K and NS8085K. Nicolas Jourdain is supported by the European Union's
Horizon 2020 research and innovation programme under grant agreements No 101003536 (ESM2025). Peter Van Katwyk was supported
by the National Science Foundation Graduate Research Fellowship Program under Grant No. 2040433. Aurélien Quiquet was funded by

the project EIS ANR-19-CE1-0015. Jonas Van Breedam and Philippe Huybrechts acknowledge support from project G091820N, funded by the Research Foundation Flanders (FWO Vlaanderen). Nicholas Golledge and Daniel Lowry were supported by the New Zealand Ministry for Business, Innovation and Employment contracts RTVU2206 ("Our Changing Coast") and ANTA1801 ("Antarctic Science Platform"). Tore Hattermann was supported by the Research Council of Norway, project number 332635. This is ISMIP6 contribution No 31. We thank Samuel Cook, Felicity McCormack and an anonymous reviewer for their suggestions that helped to improve the manuscript.

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

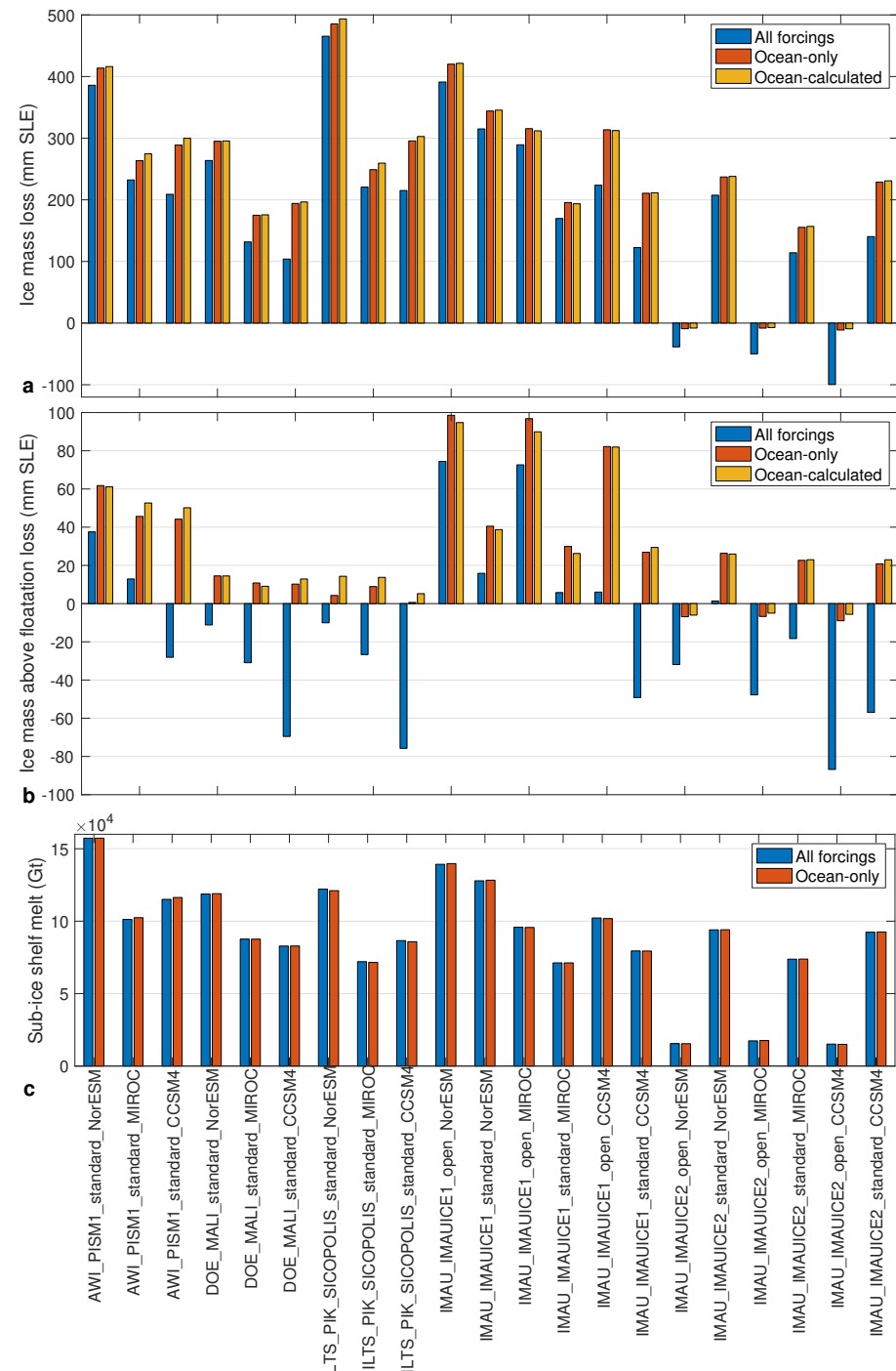

**Figure 1.** Comparison of results for experiments with ocean and atmosphere forcings (blue), experiments with ocean forcing only (ocean-only, red), and experiments with ocean and atmosphere forcings and removal of surface mass balance anomalies (ocean-calculated, yellow). Overall changes in ice volume (a, in mm SLE, positive for mass loss), ice volume above floatation (b, in mm SLE, positive for mass loss) and sub-ice shelf melt (c, in Gt, positive for ice shelf melt) during the 2015–2100 period. Results from the control run are subtracted from the experiment results in all cases.

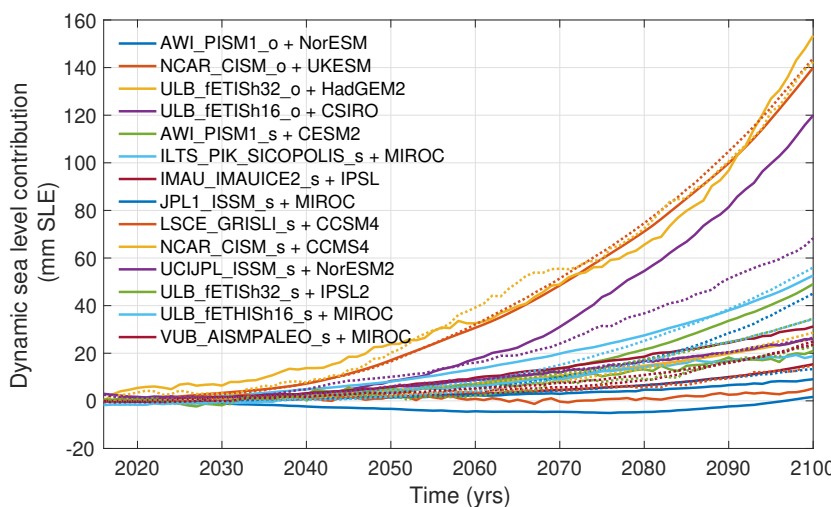

**Figure 2.** Evolution of the simulated (solid lines) and emulated (dashed lines) dynamic mass loss of the Antarctic Ice Sheet relative to the control experiment for the test set. Each color represents one simulation of the test set, with various combinations of climate model, ice sheet model and melt parameterization (*s* refers to the standard ISMIP6 melt parameterization and *o* refers to all other melt parameterizations, see Table A1).

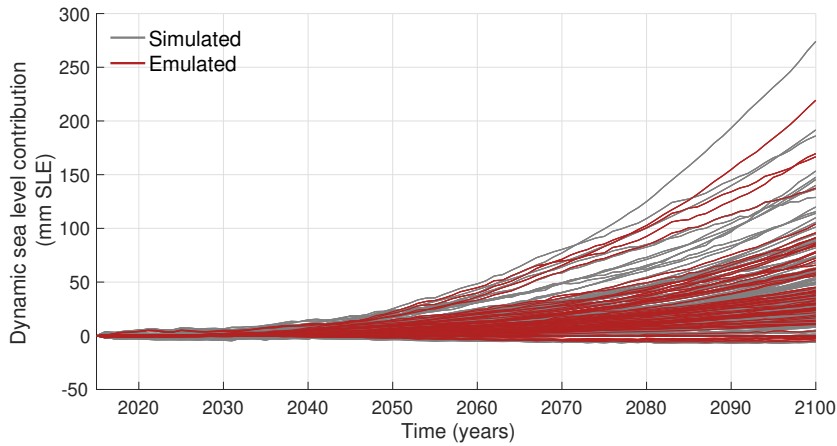

**Figure 3.** Evolution of the simulated (grey lines) and emulated (red lines) dynamic mass loss of the Antarctic Ice Sheet relative to the control experiment.

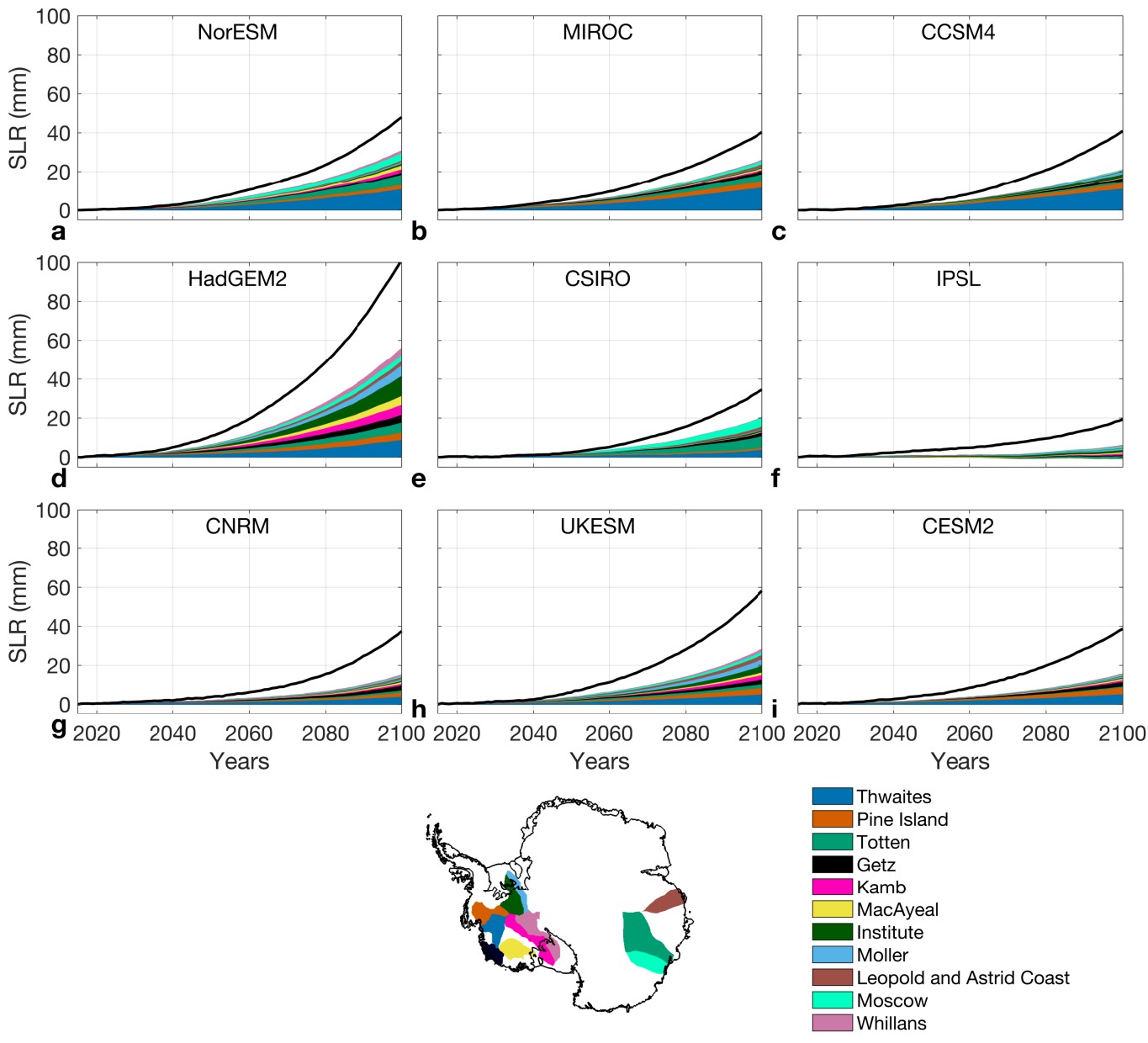

**Figure 4.** Antarctic-wide dynamic sea level contribution (black lines) and dynamic contribution from the 11 glaciers contributing most to sea level rise over the 2015–2100 period for: a) NorESM, b) MIROC, c) CCSM4, d) HadGEM2, e) CSIRO, f) IPSL, g) CNRM, h) UKESM, and i) CESM2 climate models. Values are averaged across all ice flow models and melt parameterizations. Antarctic map shows the location and names of the 11 glaciers contributing most to dynamic sea level, with colors corresponding to the different time series shown in panels a–i. Glaciers in the legend are listed by order of decreasing overall contribution to dynamic sea level contribution.

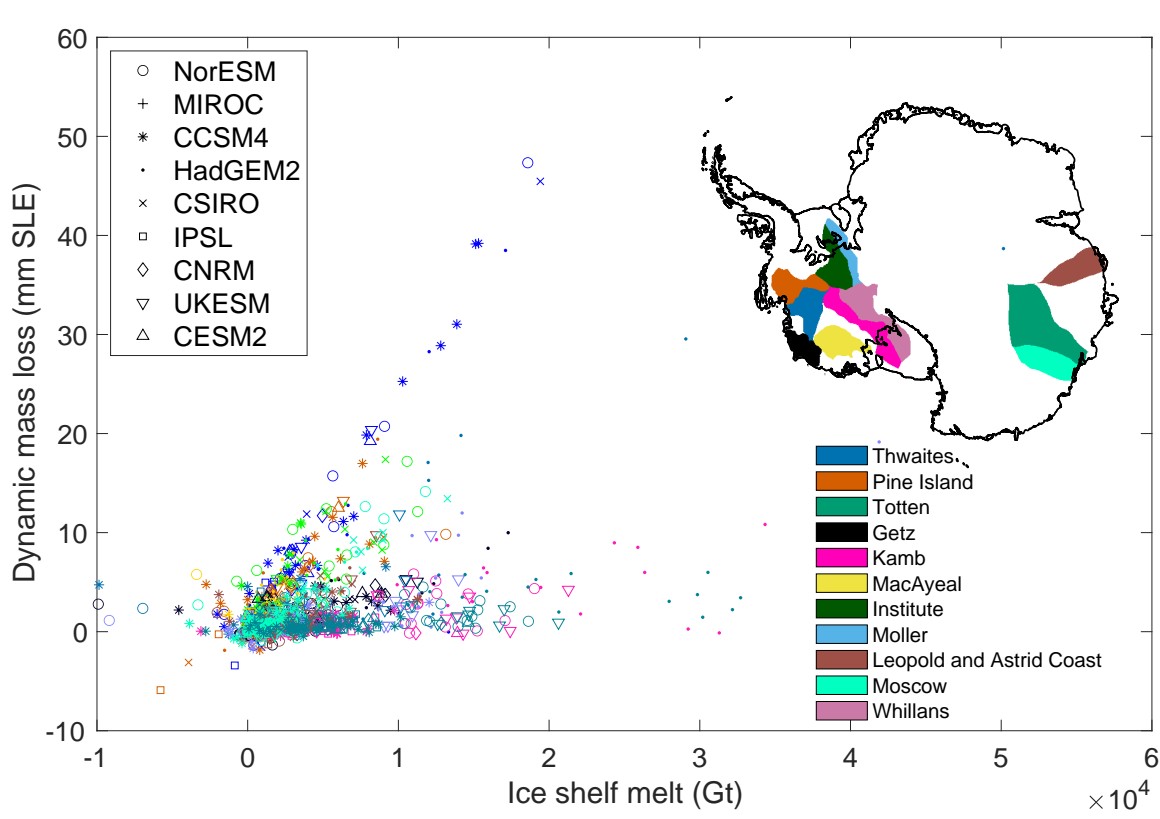

**Figure 5.** Sensitivity of dynamic mass loss to total 2015–2100 ice shelf basal melt anomaly for the 11 glaciers contributing most to Antarctic dynamic mass loss in the simulations. Colors represent the 11 different glaciers and symbols the 9 different climate models. Inset map shows the location of the 11 glaciers.

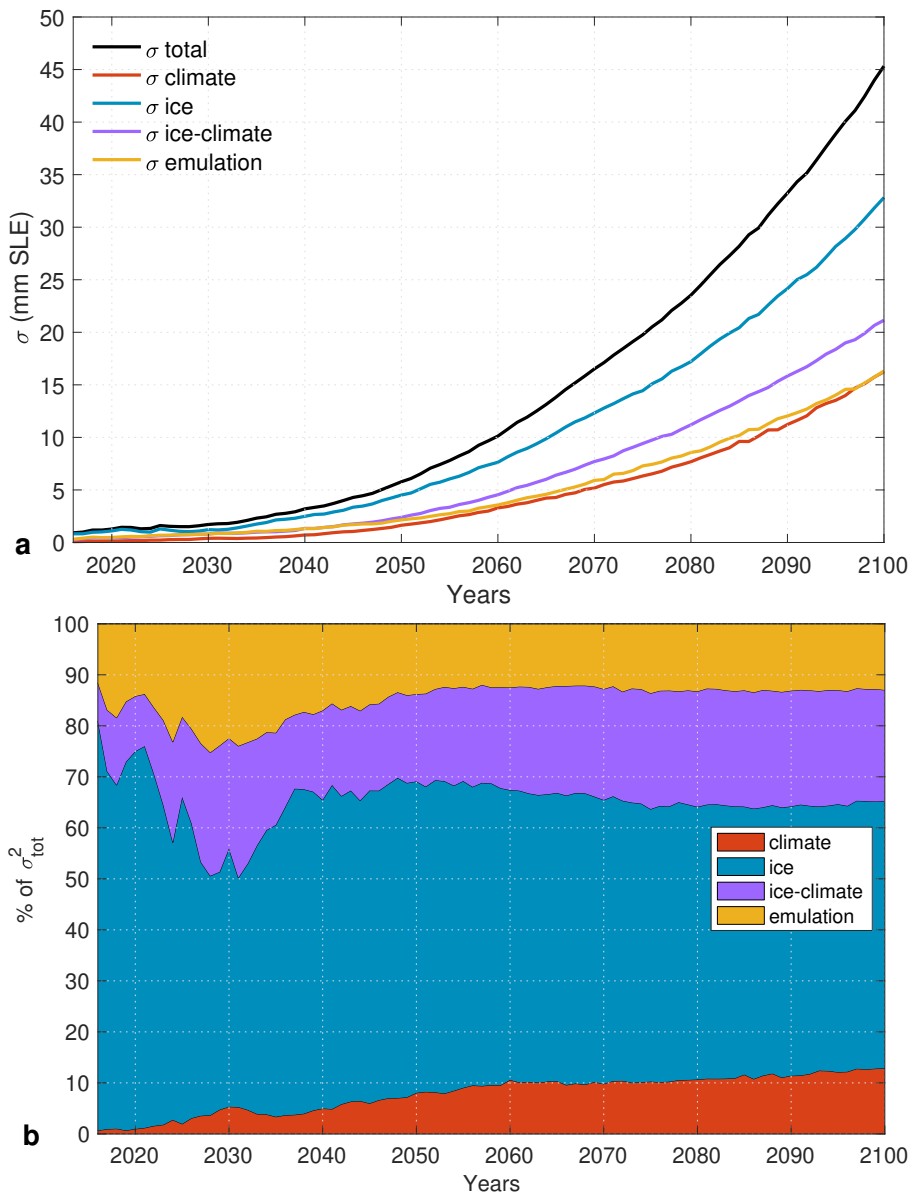

**Figure 6.** Uncertainty in dynamic mass loss of Antarctic simulations removing the trend from control run. a) Total uncertainty and uncertainty associated with the ice models, climate models, ice-climate interaction, and emulation terms. b) Relative variance of the ice models, climate models, ice-climate interaction, and emulation terms as a proportion of the total variance. The emulation term includes the emulation variance as well as the emulation-ice, the emulation-climate and the emulation-ice-climate interaction terms.

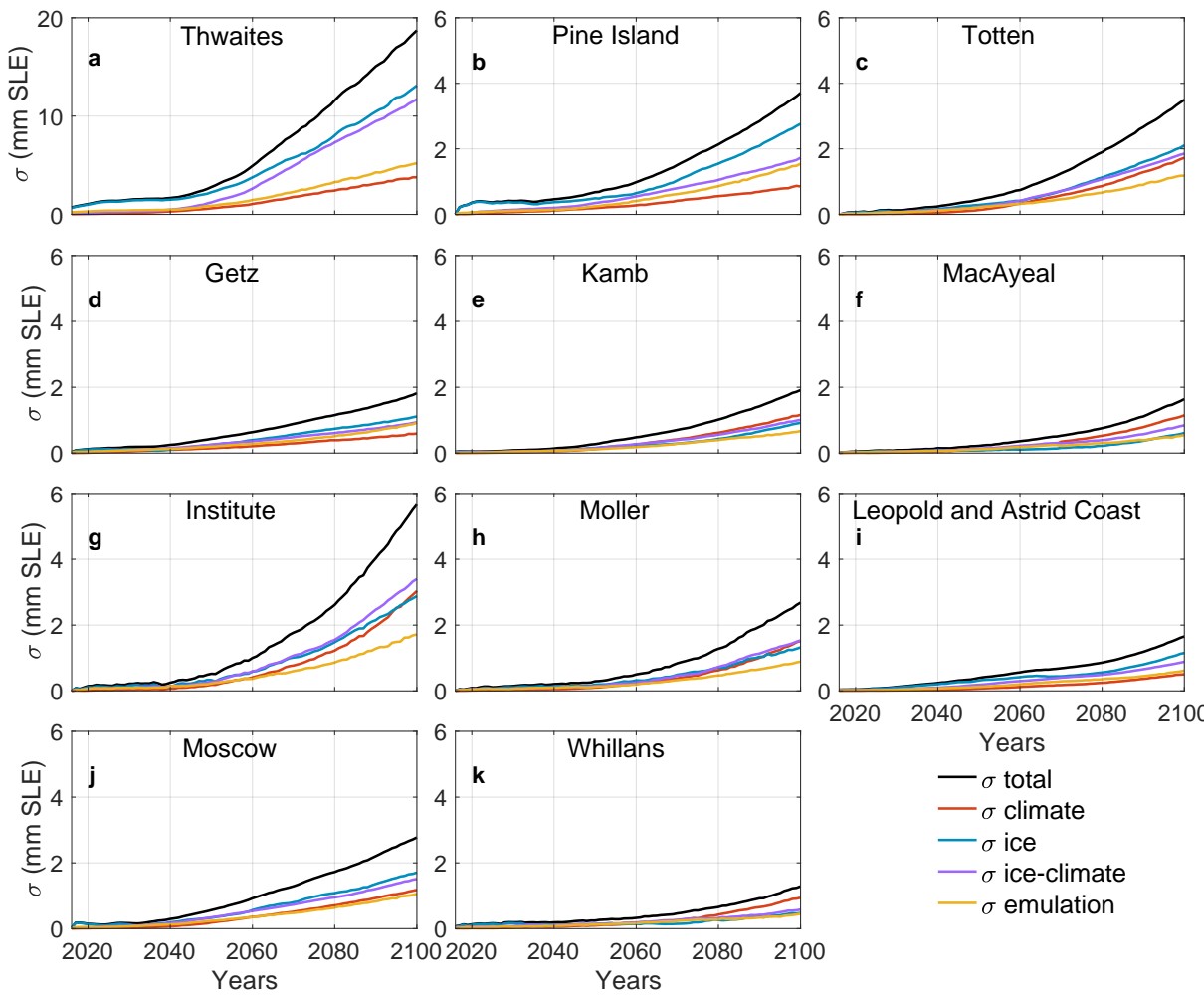

**Figure 7.** Total uncertainty and different sources of uncertainty for 11 glaciers contributing most to sea level rise: a) Thwaites, b) Pine Island, c) Totten, d) Getz, e) Kamb, f) MacAyeal, g) Institute, h) Möller, i) Leopold and Astrid Coast, j) Moscow, and k) Whillans. Note that the scale is similar for all glaciers except for Thwaites Glacier (a).

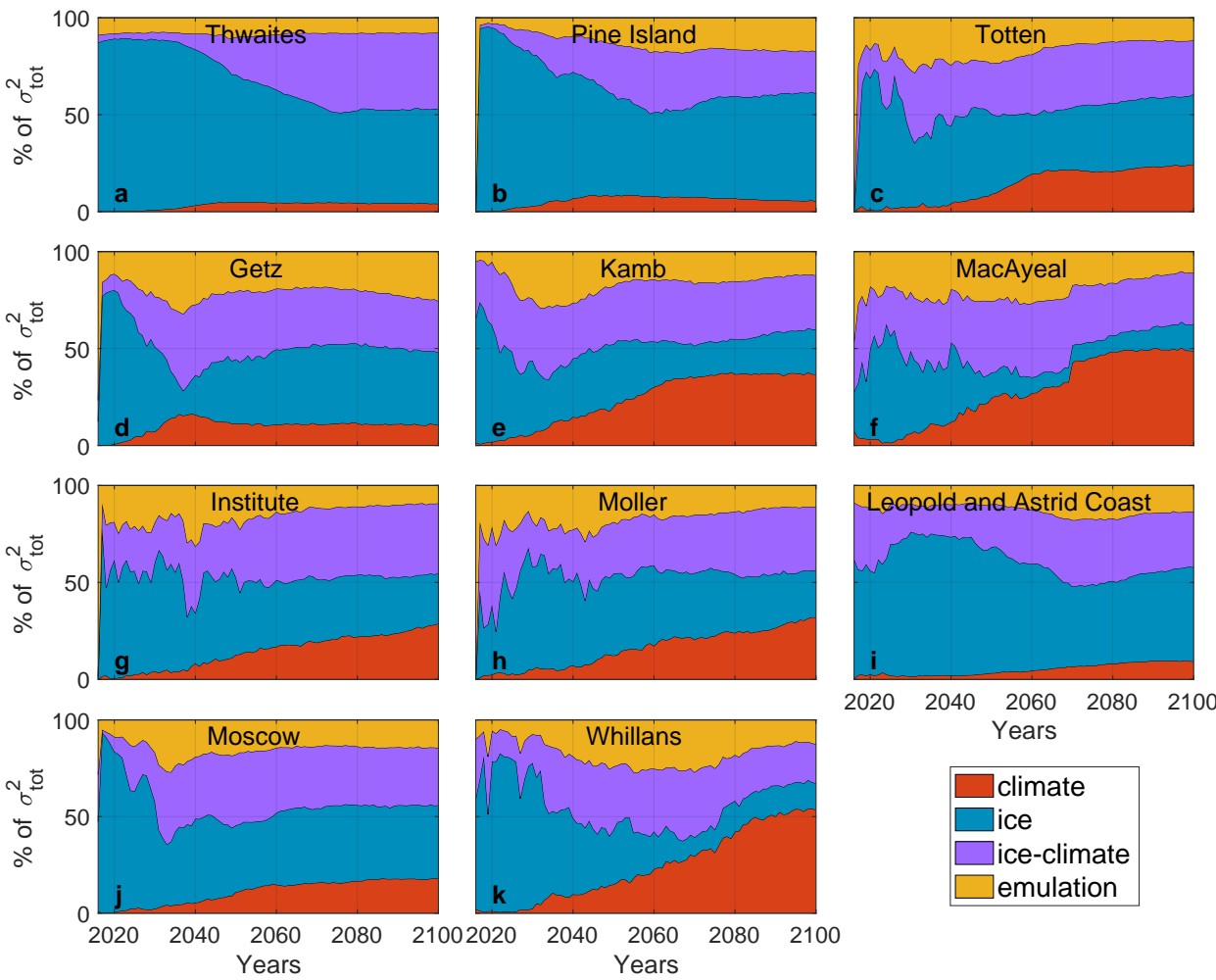

**Figure 8.** Relative variance of the ice models, climate models, ice-climate interaction, and emulation terms as a proportion of the total variance for the 2015–2100 period for: a) Thwaites, b) Pine Island, c) Totten, d) Getz, e) Kamb, f) MacAyeal, g) Institute, h) Möller, i) Leopold and Astrid Coast, j) Moscow, and k) Whillans.

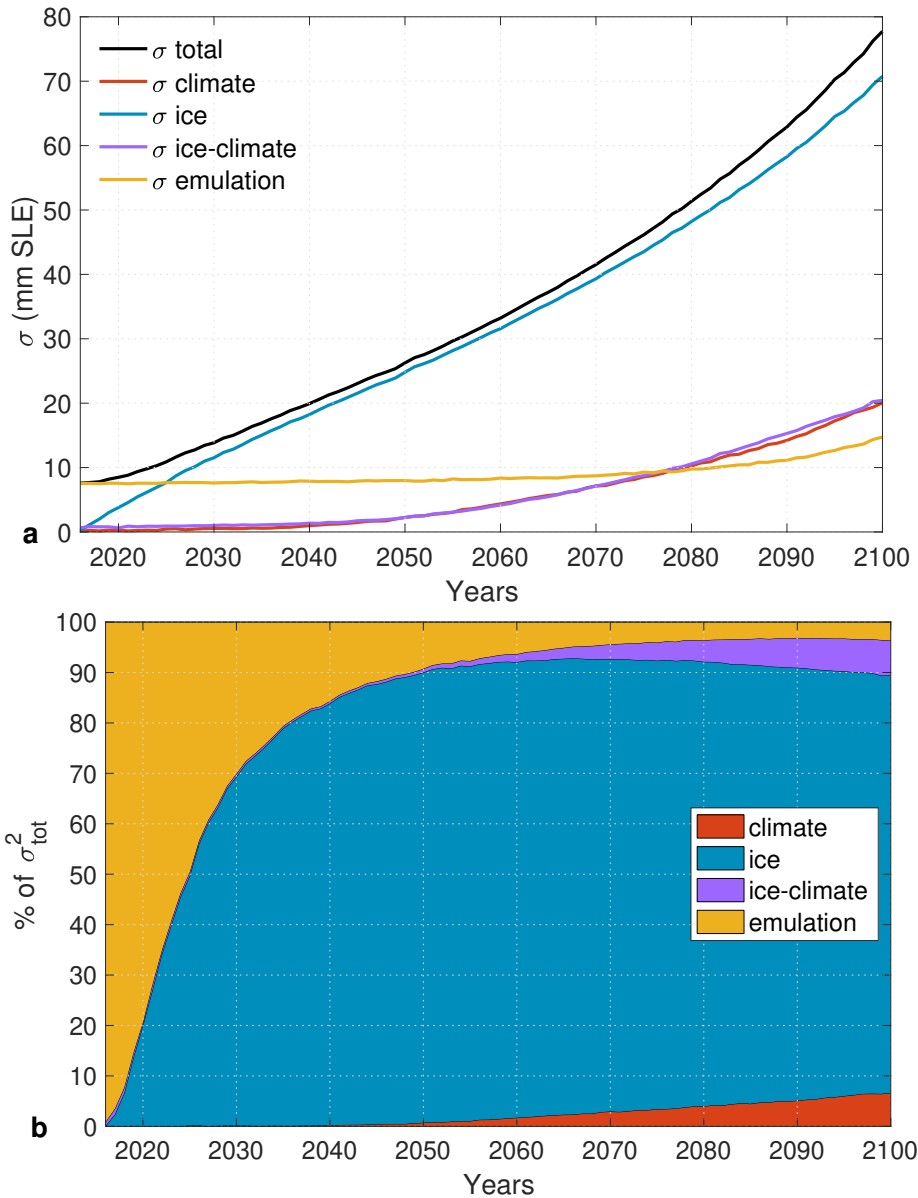

**Figure 9.** Uncertainty in dynamic mass loss of Antarctic simulations, similar to Fig. 6 but without subtracting the control run. a) Total uncertainty and uncertainty associated with the ice models, climate models, ice-climate interaction, and emulation terms. b) Relative variance of the ice models, climate models, ice-climate interaction, and emulation terms as a proportion of the total variance. The emulation term includes the emulation variance as well as the emulation-ice, the emulation-climate and the emulation-ice-climate interaction terms.