# Peer review of "Insights on the vulnerability of Antarctic glaciers from the ISMIP6 ice sheet model ensemble and associated uncertainty"

_The Cryosphere, 2023_

## Referee Comment (RC1)

**Review of Seroussi et al. (2023): Insights on the vulnerability of Antarctic glaciers from the ISMIP6 ice sheet model ensemble and associated uncertainty**

**Summary**

This paper uses the ISMIP6 ensemble of simulations to investigate the sources of uncertainty in predictions of dynamic mass loss in Antarctica, identifying those glaciers most vulnerable to dynamic changes and where most uncertainty in these predictions is coming from, both at a glacier and an ice-sheet-wide scale. As a result, the authors identify that Thwaites, Pine Island, Totten and Moscow University glaciers are the most potentially vulnerable ice masses in Antarctica, and that, generally speaking, the choice of ice-flow model is by far the largest source of uncertainty (as opposed to climate scenario, and how the climate and ice models interact) in Antarctic predictions, in contrast to those for mountain glaciers where the climate scenario dominates. Though for some glaciers (chiefly those in the Ross Sea), the climate scenario becomes dominant in the second half of the century owing to wide divergences in predicted ocean temperatures. The authors therefore conclude that the community should continue to focus on improving ice-flow models to better capture observations in order to reduce uncertainty in Antarctic mass loss predictions.

The paper is well-written and structured, with clear figures. I do not have any major concerns that require addressing, but there are a number of smaller points that could be improved to aid clarity in some places. Overall, however, this is a very good modelling paper that provides a very useful insight into which glaciers in Antarctica are most likely to suffer substantial dynamic change in the coming decades, as well as pointing the way forward to making better predictions in future.

Samuel

**Major points**
- None

**Minor points**
- p.3, l.38: 'unlike that which is observed' – the current formulation with 'what' reads a bit too informal
- p.3, l.49: 'in the overall'
- p.4, l.56: 'in the overall' – as above, one has a role in something, not to something.
- p.4, l.71: 'that which is done'
- p.4, l.72: Why only a few of the ice-flow models? I can come up with several reasons, and I imagine the correct one is 'time constraints' or 'modelling constraints', but providing a reason here would be helpful and would make the sentence read better.
- p.5, l.93: 'ice flow models'? As written, the sentence doesn't make sense, but I'm unsure if the authors are saying 'the trend in these specific ice flow models used in this study' or 'the trend in ice flow modelling generally'. Please clarify.
- p.5, l.95: Do the authors a) expect these potential non-linearities to be substantial on the timescale of this paper and b) do they see any signs of them occurring in the results?
- p.6, l.102: 'their driving stress' if the authors are referring to the glaciers mentioned earlier in the sentence or 'the ice sheet's driving stress' if the authors are referring back to the AIS in the previous sentence (and is what I think the authors mean, but I can't be sure).
- p.9, l.179: 'relative to other glaciers'
- p.9, l.180: 'that which is shown in Fig. 4'. Using 'observed' is perhaps slightly confusing when there are no observations involved.
- p.9, l.182: Similarly to the above, it's difficult to say ISMIP6 'observed' anything; consider using a different word ('that which was shown' would perhaps be best)
- p.9, l.196: I think 'similar to previous studies on mountain glaciers...or firn models' reads better
- p.9, l.199: 'Consistent with'
- p.10, l.213: 'associated with'
- p.10, l.263: 'the carbon emission scenario for'
- p.13, l.308: 'neural network emulators'

- p.13, l.326: 'that best fit observations from a large ensemble'
- Figure 2: Might it be possible to put a legend on the graph showing which colour corresponds to which simulation? Might make the figure more informative and would also fill the considerable white space at top left.
- Figure 4: Why are the glaciers in this particular order? Is it by size, contribution to mass loss or something else? This is also a different order to that given in the text in section 2.4, which the text implies (it seems to me to do so) is the order by mass loss? Please clarify.
- Figure 5: Could be worth labelling the glaciers on the inset map, just so readers don't have to check back to Figure 4 to remind themselves which one's which, if they're not familiar with their locations
- Figure 6: Caption, 'associated with', and 'as a proportion of the total variance' (?)
- Figure 8: Caption, 'Relative contribution...to the total variance' (?)
- Figure 9: Caption, 'associated with', 'Relative contribution...to the total variance'

---

## Author Comment (AC1)

**Response to reviewers**

**Reviewer 1**

**Summary**
This paper uses the ISMIP6 ensemble of simulations to investigate the sources of uncertainty in predictions of dynamic mass loss in Antarctica, identifying those glaciers most vulnerable to dynamic changes and where most uncertainty in these predictions is coming from, both at a glacier and an ice-sheet-wide scale. As a result, the authors identify that Thwaites, Pine Island, Totten and Moscow University glaciers are the most potentially vulnerable ice masses in Antarctica, and that, generally speaking, the choice of ice-flow model is by far the largest source of uncertainty (as opposed to climate scenario, and how the climate and ice models interact) in Antarctic predictions, in contrast to those for mountain glaciers where the climate scenario dominates. Though for some glaciers (chiefly those in the Ross Sea), the climate scenario becomes dominant in the second half of the century owing to wide divergences in predicted ocean temperatures. The authors therefore conclude that the community should continue to focus on improving ice-flow models to better capture observations in order to reduce uncertainty in Antarctic mass loss predictions.

The paper is well-written and structured, with clear figures. I do not have any major concerns that require addressing, but there are a number of smaller points that could be improved to aid clarity in some places. Overall, however, this is a very good modelling paper that provides a very useful insight into which glaciers in Antarctica are most likely to suffer substantial dynamic change in the coming decades, as well as pointing the way forward to making better predictions in future.
Samuel

We thank the reviewer for his careful reading and his suggestions to improve the clarity of the paper. We have included these suggestions and provide a point-by-point response below.

**Major points**
• None

**Minor points**
• p.3, l.38: 'unlike that which is observed' – the current formulation with 'what' reads a bit too informal
Done

• p.3, l.49: 'in the overall'
Done

• p.4, l.56: 'in the overall' – as above, one has a role in something, not to something.
Done

- p.4, l.71: 'that which is done'

Done

- p.4, l.72: Why only a few of the ice-flow models? I can come up with several reasons, and I imagine the correct one is 'time constraints' or 'modelling constraints', but providing a reason here would be helpful and would make the sentence read better.

Thanks for the suggestion. These ocean-only experiments were part of a lower tier of experiments, and therefore only models interested specifically in the response to ocean changes performed them. We added this explanation in the text.

- p.5, l.93: 'ice flow models'? As written, the sentence doesn't make sense, but I'm unsure if the authors are saying 'the trend in these specific ice flow models used in this study' or 'the trend in ice flow modelling generally'. Please clarify.

We remove the trend associated with each model individually in order to get the response to climate change associate to each model. This problem comes from the limited ability of ice flow models to accurately capture the recent changes, but the way in which they respond varies from one model to the next, so we need to remove the trend associated to each model. We clarified the text.

- p.5, l.95: Do the authors a) expect these potential non-linearities to be substantial on the timescale of this paper and b) do they see any signs of them occurring in the results?

The extent of the non-linearities will vary from one model to the next and are difficult to estimate without additional experiments. Unfortunately there is not much information to inform if this is happening in the results and it would be pure speculation to do so, therefore we do not discuss this question in the current manuscript.

- p.6, l.102: 'their driving stress' if the authors are referring to the glaciers mentioned earlier in the sentence or 'the ice sheet's driving stress' if the authors are referring back to the AIS in the previous sentence (and is what I think the authors mean, but I can't be sure).

Thanks for the suggestions to clarify. We do refer to the glaciers mentioned earlier.

- p.9, l.179: 'relative to other glaciers'

Done

- p.9, l.180: 'that which is shown in Fig. 4'. Using 'observed' is perhaps slightly confusing when there are no observations involved.

Done

- p.9, l.182: Similarly to the above, it's difficult to say ISMIP6 'observed' anything; consider using a different word ('that which was shown' would perhaps be best)

Done

- p.9, l.196: I think 'similar to previous studies on mountain glaciers...or firn models' reads better

Done

- p.9, l.199: 'Consistent with'

Done

- p.10, l.213: 'associated with'

Done

- p.10, l.263: 'the carbon emission scenario for'

Done

- p.13, l.308: 'neural network emulators'

Done

- p.13, l.326: 'that best fit observations from a large ensemble'

Done

- Figure 2: Might it be possible to put a legend on the graph showing which colour corresponds to which simulation? Might make the figure more informative and would also fill the considerable white space at top left.

The exact experiment with the combination of ice, climate and melt parameterization used in the test set was added in the legend, as suggested by both reviewers and the editor. See new legend and caption in Figure 2.

- Figure 4: Why are the glaciers in this particular order? Is it by size, contribution to mass loss or something else? This is also a different order to that given in the text in section 2.4, which the text implies (it seems to me to do so) is the order by mass loss? Please clarify.

The glaciers are indeed ranked in the order of the ones contributing most to changes on average for all the climate models. In the text the order can sometimes be different since they are sometimes listed by regions first, or by sensitivity. We clarified the order in the figure caption.

- Figure 5: Could be worth labelling the glaciers on the inset map, just so readers don't have to check back to Figure 4 to remind themselves which one's which, if they're not familiar with their locations

We added a second legend with the colors and names of the glaciers similar to Figure 4.

- Figure 6: Caption, 'associated with', and 'as a proportion of the total variance' (?)

Done

- Figure 8: Caption, 'Relative contribution...to the total variance' (?)

Done

- Figure 9: Caption, 'associated with', 'Relative contribution...to the total variance'

Done

**Reviewer 2**

**Summary**
In this paper, the authors quantify projections of mass loss from individual glaciers around the Antarctic Ice Sheet out to 2100, with the goal of (1) identifying the glaciers most vulnerable to significant mass loss, and (2) quantifying the dominant sources of uncertainty in these projections. The authors identify Thwaites Glacier, Pine Island Glacier, Totten Glacier, and Moscow University Glacier as being the regions most likely to experience significant mass loss in the next century, and they identify the significant role that the choice of ice flow model has in contribution to uncertainty. I found the paper to be clear and the experiments themselves to be well structured and explained. The figures were easy to read and the structure of the paper highlighted the takeaways well. Below I outline some areas for potential further detail, which may allow the reader to understand the underlying assumptions of the study better.

We thank the reviewer for their constructive comments and the suggestions to make the underlying assumptions of the manuscript easier to follow. We have included these suggestions and provide a point-by-point response below.

- **Uncertainty Study:** while the explanation of the uncertainty results was quite clear, I found the description of the methodology to be a bit sparse, which made it difficult to understand the study itself. The authors state that they use ANOVA to partition the uncertainty into individual contributions. For those unfamiliar (or less familiar) with these statistical methods, a few more sentences about what specific test was used and what the method entails would be valuable. For example, are there any underlying assumptions about the probability distributions involved? Are the varying melt parameterizations used in the ice flow models considered part of the ice flow model variance, the climate model variance, or the interaction between the two?
  We added additional information about ANOVA and how it is used in our approach and how the ice-ocean term captures the interaction between the two variables ice and climate in section 3.2.

- **Melt Parameterizations:** given that the focus of the study is the effect of melt forcing, it would be useful to describe the various melt parameterizations used in the ice flow models, as I imagine this significantly affects the response of the glaciers to climate forcing.
  We added a short description of the ISMIP6 and other main melt parameterizations used in different ice models in section 2.1

- **Ice Flow Models:** Tables 1 and 2 do a good job of outlining the experiments and the various climate and ice flow models used. However, it would be valuable to have a small description of the ways the ice flow models differ themselves, as this would provide some context to the result that most of the posterior uncertainty is due to uncertainty in the choice of ice flow model. A similar table that describes the pieces of each ice flow model, the assumptions they used (full Stokes vs. SIA, basal sliding parameterization, temperature-dependent rheology) would allow for a bit more insight into this result.

This information is similar to Table 3 in Seroussi et al. (2020), so we added this table as an Appendix to this manuscript for readers to easily find the main model characteristics and referenced it throughout the text.

- **Figures:** A few of the figures could use some minor adjustments to make them clearer:
  - o Fig 4: the axis labels and titles are hard to read; increase in font size would help significantly

This figure does need a larger font size and consolidated axis. We updated it with font size similar to other figures to make it easier to read.

  - o Fig 3: making the lines in the legend thicker would help readability

We updated the legend in Fig.3 to make the lines easier to distinguish.

  - o Fig 2: is it valuable to have a legend so the reader can identify which simulation produces the most/least sea level contributions?

Both reviewers and the editor were interested in knowing more about the simulations used, so we added a legend with the information providing the ice and climate model configuration on the figure.